# Reasoning-SQL: Reinforcement Learning with Partial Rewards for Reasoning-Enhanced Text-to-SQL

**Mohammadreza Pourreza***
Google Cloud
pourreza@google.com

**Shayan Talaei***
Stanford University
stalaei@stanford.edu

**Ruoxi Sun**
Google DeepMind
ruoxis@google.com

**Xingchen Wan**
Google Cloud
xingchenw@google.com

**Hailong Li**
Google Cloud
hailongli@google.com

**Azalia Mirhoseini**
Stanford University
azalia@stanford.edu

**Amin Saberi**
Stanford University
saberi@stanford.edu

**Sercan Ö. Arık**
Google Cloud
soarik@google.com

## Abstract

Text-to-SQL is a challenging task involving multiple reasoning-intensive subtasks, including natural language understanding, database schema comprehension, and precise SQL query formulation. Existing approaches often rely on handcrafted reasoning paths with inductive biases that can limit their overall effectiveness. Motivated by the recent success of reasoning-enhanced models such as DeepSeek R1 (Guo et al., 2025) and OpenAI o1 (Jaech et al., 2024), which effectively leverage reward-driven self-exploration to enhance reasoning capabilities and generalization, we propose a novel set of partial rewards tailored specifically for the Text-to-SQL task. Our reward set includes schema-linking, AI feedback, N-gram similarity, and syntax check, explicitly designed to address the reward sparsity issue prevalent in reinforcement learning (RL). Leveraging group relative policy optimization (GRPO), our approach explicitly encourages large language models (LLMs) to develop intrinsic reasoning skills necessary for accurate SQL query generation. With models of different sizes, we demonstrate that RL-only training with our proposed rewards consistently achieves higher accuracy and superior generalization compared to supervised fine-tuning (SFT). Remarkably, our RL-trained 14B-parameter model significantly outperforms larger proprietary models, e.g. o3-mini by 4% and Gemini-1.5-Pro-002 by 3% on the BIRD benchmark. These highlight the efficacy of our proposed RL-training framework with partial rewards for enhancing both Text-to-SQL accuracy and reasoning capabilities.

## 1 Introduction

Bridging the gap between human language and database queries, Text-to-SQL focuses on translating natural language questions into executable SQL. The advent of large language models (LLMs) has yielded significant progress, with most state-of-the-art Text-to-SQL systems now relying on LLMs for synthesizing SQL queries. However, the performance of these methods is inherently limited by the reasoning capabilities of the underlying models, making it essential to explicitly enhance reasoning in the Text-to-SQL domain. Accurate SQL generation requires both deep understanding of natural language and sophisticated reasoning over complex database schemas (Cao et al., 2021; Talaei et al., 2024; Pourreza & Rafiei, 2024a). Conventional approaches for LLM adaptation, predominantly based

---

*Equal contribution.

on supervised fine-tuning (SFT) (Pourreza & Rafiei, 2024b; Li et al., 2024b) or few-shot prompting (Gao et al., 2023; Pourreza & Rafiei, 2024a), often fall short when the task demands extended reasoning, especially in cases of ambiguous or multi-step queries.

Recent advances in large reasoning models, such as OpenAI's o1 (Jaech et al., 2024) and DeepSeek-R1 (Guo et al., 2025), have shown that additional compute during inference to produce judicious reasoning trajectories can significantly improve response quality. Motivated by these, we propose Reasoning-SQL, a learning framework with reinforcement learning (RL) to enhance the reasoning process of LLMs for Text-to-SQL by encouraging the generation of detailed intermediate reasoning steps that ultimately lead to more accurate SQL queries.

A core component of RL-based training is the reward function, which is crucial for guiding the model. For Text-to-SQL, the most intuitive reward would be the execution accuracy. However, its binary and sparse nature provides limited feedback when the model partially captures correct logic or schema relationships (Nguyen et al., 2025). Reward sparsity is a notable challenge in many RL-based training approaches, often impeding the optimization of policy models (Blier & Ollivier, 2021; Rengarajan et al., 2022; Jaderberg et al., 2016). To address this, we design a composite reward that integrates multiple partial rewards to address the reward sparsity problem for Text-to-SQL, namely LLM-as-a-Judge, Syntax Check, Schema Linking, and N-gram Similarity rewards. We employ Group Relative Policy Optimization (GRPO) (Shao et al., 2024) to effectively integrate these reward signals. By generating multiple candidate queries per input and evaluating them relative to each other, our approach provides a robust and informative feedback mechanism that directly optimizes both the intermediate reasoning process and the final execution accuracy.

Our extensive ablation studies on various reward designs underscore the effectiveness of our proposed partial rewards in boosting model performance beyond what execution accuracy alone can achieve. We validate Reasoning-SQL on several challenging benchmarks, including BIRD, Spider, Spider-DK, and Spider-Syn (Li et al., 2024c), demonstrating that smaller models trained with our approach not only outperform conventional SFT methods but also exceed much larger proprietary base models, with gains of 4% over o3-mini and 3% over Gemini-1.5-Pro-002. Moreover, our models integrated into standard text-to-SQL pipelines deliver a state-of-the-art 72.78% execution accuracy, all at a 93% lower monetary inference cost. We further establish the superiority of the model's emergent reasoning capabilities by showing that its naturally evolved, structured reasoning outperforms well-designed, hand-crafted step-by-step approaches.

Our main contributions can be summarized as: (1) Automatic RL-based Reasoning Optimization: We introduce, Reasoning-SQL, the first RL-based framework that automatically optimizes the reasoning process of LLMs for the Text-to-SQL task. (2) Novel Partial Rewards Suite: We propose a novel suite of partial rewards specifically designed for Text-to-SQL, with comprehensive ablation studies demonstrating their effectiveness. (3) State-of-the-Art and Cost-Effective Performance: Our method achieves state-of-the-art results on the BIRD test set among the best open-source models, competing with significantly larger proprietary models while being much more cost effective.

## 2  Related Works

**Text-to-SQL.** Early Text-to-SQL methods used rule-based and template-driven approaches with handcrafted grammars and keyword matching (Androutsopoulos et al., 1995; Hristidis et al., 2003; Li & Jagadish, 2014; Popescu et al., 2003; 2004). With deep learning, the focus shifted to neural semantic parsing using sequence-to-sequence models like Seq2SQL (Zhong et al., 2017) and SQLNet (Xu et al., 2017), which employed reinforcement learning to boost execution accuracy (Zhong et al., 2017; Xu et al., 2017; Popescu et al., 2004; Li & Jagadish, 2014). Later work further advanced these techniques by integrating schema linking, structured decoding with graph-based encoders, and constrained generation (e.g., IRNet (Guo et al., 2019) and RAT-SQL (Wang et al., 2019; Cai et al., 2021)), along with incremental parsing strategies such as PICARD to ensure syntactic validity (Scholak et al., 2021; Xu et al., 2017; Popescu et al., 2004; Li & Jagadish, 2014). LLMs have revolutionized Text-to-

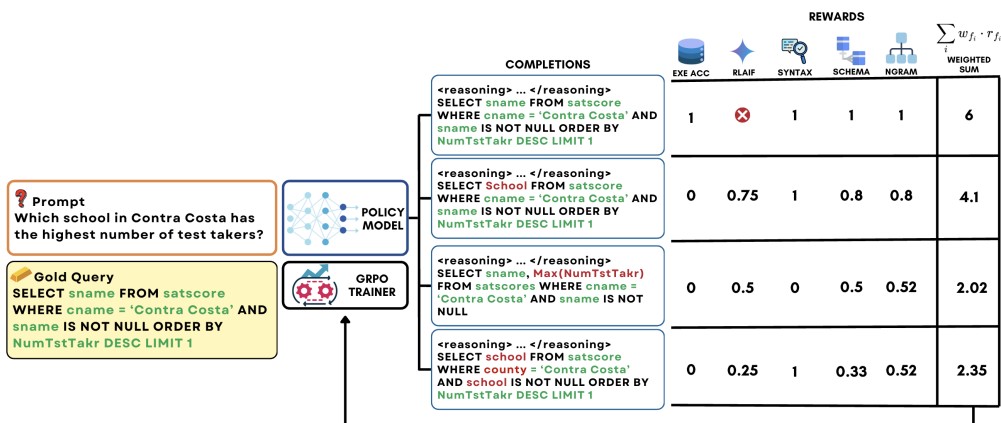

Figure 1: Overview of the GRPO-based Text-to-SQL training pipeline. For each natural language prompt q and its associated database schema, the policy model $\pi_\theta$ generates a group of candidate SQL queries. Each candidate is evaluated using a suite of reward functions to produce a composite reward. These rewards are then used to compute advantages and update the policy via GRPO.

SQL by enabling zero-shot and few-shot learning through advanced prompt engineering and in-context learning (Brown et al., 2020; Wei et al., 2022; Dong et al., 2023; Pourreza & Rafiei, 2024a; Gao et al., 2023). Recent methods use multi-step pipelines—integrating schema linking, self-correction, and execution feedback—to iteratively refine SQL generation (Pourreza et al., 2024; Wang et al., 2023; Sun et al., 2023a; Xie et al., 2025; Gao et al., 2024; Talaei et al., 2024), while other approaches directly fine-tune open-source LLMs on large Text-to-SQL corpora (Li et al., 2024b;c;a; Nan et al., 2023; Pourreza & Rafiei, 2024b). Although reinforcement learning was initially employed to improve SQL generation by leveraging execution feedback (Zhong et al., 2017; Xu et al., 2017; Popescu et al., 2004; Li & Jagadish, 2014), more recent RL-based methods that incorporate intermediate reward signals and chain-of-thought prompting (Nguyen et al., 2025; Zelikman et al., 2022; Jaderberg et al., 2016) have only yielded modest improvements compared to techniques that directly capitalize on the reasoning capabilities of LLMs (Kojima et al., 2022; Sun et al., 2023b).

**RL for reasoning.** OpenAI's O-series models achieve strong performance in complex reasoning tasks, such as mathematical problem-solving, code generation, and logical deduction, primarily due to their use of chain-of-thought prompting combined with RL training (Jaech et al., 2024; Wei et al., 2022). Building on these successes, the research community has worked to reproduce and extend these capabilities (Qin et al., 2024; Zhao et al., 2024). Notably, DeepSeek R1 (Guo et al., 2025) and Kimi-K1.5 (Team et al., 2025) have emerged as strong contenders, leveraging RL-based training to foster autonomous internal reasoning and reflection (Shao et al., 2024). This progress has spurred further work in enhancing performance on reasoning-intensive tasks such as code generation (Zeng et al., 2025), software evolution (Wei et al., 2025), mathematical reasoning (Luo et al., 2025), and visual maze solving (Dao & Vu, 2025). Despite Text-to-SQL requiring similarly sophisticated reasoning, involving user intent interpretation, schema comprehension, and precise SQL query construction, there remains a notable gap in explicitly applying RL-based training methods to improve LLM reasoning in this domain.

## 3 Methodology

To train a reasoning language model for Text-to-SQL, we employ an RL framework built upon GRPO and complemented by a set of carefully crafted reward functions. This combination is specifically designed to provide task-specific guidance, addressing key hurdles in Text-to-SQL generation, including accurate schema linking, the generation of valid SQL syntax, and the limitations posed by sparse execution feedback. An overview of our training flow is presented in Figure 1.

## 3.1 Reinforcement Learning Protocol

For RL fine-tuning of LLMs, methods based on policy optimization, such as Proximal Policy Optimization (PPO) (Schulman et al., 2017) and GRPO (Shao et al., 2024), have been explored. Given the demonstrated effectiveness of GRPO in training models like R1 (Guo et al., 2025) and its advantages over PPO, including eliminating the need for a separate value model and reducing memory requirements, we focus on GRPO to effectively optimize the policy model $\pi_\theta$ for SQL generation. For each input, consisting of a natural language question $q$ and its associated database schema, the model generates a group of $G$ candidate SQL queries, $\{o_1, o_2, \ldots, o_G\}$. Each candidate is evaluated using a composite reward function that is designed to capture both the end-goal of correct execution and several intermediate quality measures essential to Text-to-SQL.

GRPO leverages the relative performance of candidates within the group to compute an advantage $A_i$ for each output, guiding policy updates according to the following objective:

$$J_{\text{GRPO}}(\theta) = \mathbb{E} \left[ \frac{1}{G} \sum_{i=1}^{G} \min \left( \frac{\pi_\theta(o_i|q)}{\pi_{\theta_{\text{old}}}(o_i|q)} A_i, \text{clip} \left( \frac{\pi_\theta(o_i|q)}{\pi_{\theta_{\text{old}}}(o_i|q)}, 1 - \epsilon, 1 + \epsilon \right) A_i \right) \right]$$
$$- \beta D_{\text{KL}}(\pi_\theta \| \pi_{\text{ref}}), \tag{1}$$

where $\pi_{\theta_{\text{old}}}$ is the policy before the update, $\pi_{\text{ref}}$ is the reference policy (typically the initial model), and $\epsilon$ and $\beta$ are hyperparameters controlling the update step and divergence regularization. By generating multiple candidates per input, GRPO naturally accommodates the inherent ambiguities and challenges of mapping natural language to SQL queries, ensuring that feedback is both robust and informative.

## 3.2 Reward Design

Having a well-shaped reward is key to the efficacy of RL training (Akalin & Loutfi, 2021; Bouktif et al., 2023; Trella et al., 2023). For Text-to-SQL, the most intuitive reward would be the execution accuracy (Nguyen et al., 2025; Li et al., 2024c), which can check if the generated SQL queries produce the expected output after getting executed, commonly known as Reinforcement Learning from Execution Feedback (RLEF). However, the binary and sparse nature of it would create challenges for RL optimization (Blier & Ollivier, 2021; Rengarajan et al., 2022; Jaderberg et al., 2016), as many candidates receive the same feedback despite varying degrees of correctness, and the infrequent reward signals offer limited guidance for differentiating between nearly correct and completely incorrect outputs. To mitigate this, we introduce a set of partial rewards that provide more granular guidance, such as LLM-as-a-judge reward, commonly referred to as Reinforcement Learning from AI Feedback (RLAIF). The final reward used to optimize the policy model is the weighted sum of all partial and the execution accuracy rewards. Figure 2 provides an example of how each of the rewards is calculated given the gold (ground truth) and a generated query.

**Execution accuracy reward (RLEF).** This reward directly assesses whether the generated SQL query, when executed against the database, produces the correct results. For each candidate query, we perform an execution alongside the ground-truth query and compare their resulting outputs, with the perfect match earning the full reward. While execution accuracy serves as the ultimate benchmark aligned with the downstream goals, its binary nature results in a sparse reward signal. This sparsity is a limitation, as many queries might contain elements of the correct logic but still fail to yield the exact correct output, thus receiving no reward.

**LLM-as-a-judge reward (RLAIF).** To supplement binary execution feedback, we employ an LLM as a judge. Using a specially designed prompt (provided in Appendix Section A.3) that incorporates specific rubrics for comparing SQL queries, the LLM evaluates candidate queries against the ground truth. This evaluation is performed exclusively for queries with zero execution accuracy, assessing them based on criteria such as logical consistency, structural similarity, and semantic correctness. This provides a nuanced partial reward, differentiating between various incorrect answers. The ✗ in Figure 1 indicates a correct query, thus requiring no additional AI feedback.

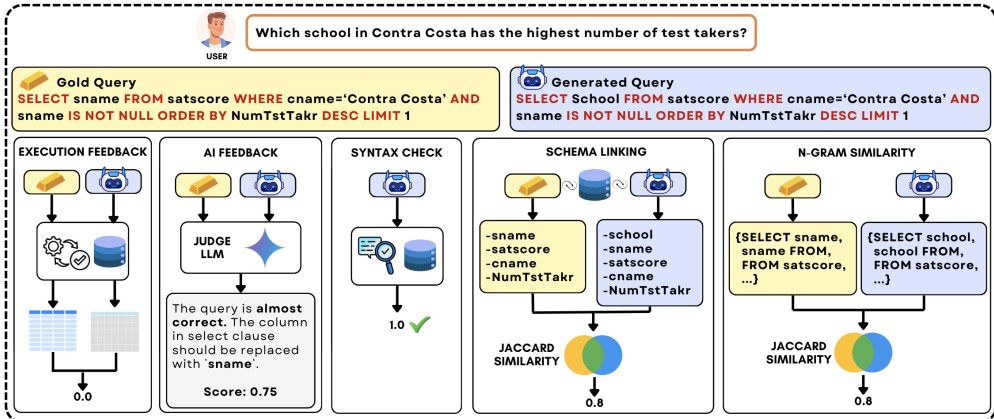

Figure 2: Example partial reward calculation for a generated SQL query, illustrating how each reward component, execution accuracy, llm-as-a-judge (AI Feedback), syntax check, schema linking, and N-gram similarity, is derived by comparing the candidate query with the gold (ground truth) query.

**Syntax check reward.** Syntactic validity is a fundamental prerequisite for any query to be considered meaningful or executable. We design a corresponding reward so that a positive score is assigned when the generated SQL query is syntactically valid and executes without errors, regardless of whether it returns the exact correct result. This reward strategy aids in distinguishing between incorrect queries by rewarding syntactic correctness, thereby instructing the model to prioritize well-formed query generation.

**Schema linking reward.** Accurate schema linking is critical in Text-to-SQL (Wang et al., 2019; Pourreza & Rafiei, 2024a; Talaei et al., 2024; Caferoğlu & Ulusoy, 2024), as the model must correctly identify and reference the relevant tables and columns. Our schema linking reward quantifies the Jaccard similarity between the set of used schema items in the candidate query relative to the gold query, directly addressing the challenge of mapping natural language entities to the correct parts of the database schema. Figure 2 illustrates an example of schema linking reward computation for a pair of predicted query and gold query.

**N-gram similarity reward.** The structured nature of SQL allows leveraging token-level overlap as a measure of similarity. The N-gram similarity reward computes the Jaccard similarity between N-grams in the candidate and gold SQL queries. Given SQL's inherent hierarchical syntax and structured format, this reward promotes alignment with the correct query's lexical and syntactic structure, even when minor differences exist. Figure 2 illustrates the computation of N-gram similarity using bigrams (N=2), as employed in this paper.

**Format reward.** Finally, we include a format reward that encourages adherence to a predefined output structure (e.g., using <reasoning> and <answer> tags). Outputs that conform to this pattern receive a reward boost, thereby enhancing clarity and consistency. This structure also triggers zero-shot chain-of-thought reasoning in the policy model, which progressively improves as training advances to optimize for the reward.

Together, the weighted sum of these evaluation functions constitutes the final composite reward. Formally, given evaluation functions $f_i$, each providing a reward $r_{f_i}$, and corresponding weights $w_{f_i}$, the final composite reward is defined as: $r = \sum_i w_{f_i} \cdot r_{f_i}$. In our experiments, the weights are carefully chosen to ensure that no incorrect SQL query can achieve a higher overall reward than a correct query. This reward aggregation yields dense, informative feedback, significantly enhancing the efficacy of the GRPO training framework to guide the model toward generating SQL queries that are both syntactically valid and semantically accurate (Section 5.1). Figure 1 illustrates the overall training pipeline where candidate queries are generated, evaluated through the specialized reward functions, and reinforced the model's reasoning policy $\pi_\theta$ towards generating more accurate SQL queries. For the pseudo-code of our algorithm please refer to Appendix A.7.

### 3.3 Supervised Fine-Tuning Baselines

To establish robust baselines, we implement two SFT approaches: direct SQL prediction and the bootstrapped STaR-SFT method (Zelikman et al., 2022). In the direct SFT paradigm, the model is trained via maximum likelihood estimation on gold SQL queries, serving as a reliable benchmark. To assess the benefits of training on reasoning traces with SFT, we introduce a second baseline—STaR-SFT—based on the Self-Taught Reasoner (STaR) framework (Zelikman et al., 2022). The STaR-SFT method explicitly integrates chain-of-thought (CoT) reasoning by leveraging the Divide-and-Conquer prompt from CHASE-SQL (Pourreza et al., 2024) and the Gemini-1.5-Pro-002 model to generate CoT traces (Wei et al., 2022). When these CoT sequences yield correct SQL outputs, the corresponding samples are added to the training set; otherwise, the ground truth SQL is provided as a hint to guide the model toward producing accurate CoT chains.

## 4 Experimental Setup

In our experiments, we are going to address the following research questions:

**RQ1) Training Objective Ablation**: How do different reward functions and the inclusion of reasoning affect performance relative to SFT? **RQ2) Baseline Comparison**: How does Reasoning-SQL compare to existing Text-to-SQL methods in execution accuracy and schema linking? **RQ3) Emergent Reasoning**: How does structured reasoning emerge during RL training, and how does it compare with human-designed strategies? **RQ4) Generalization Analysis**: How well does Reasoning-SQL generalize to out-of-distribution SQL queries and related tasks?

### 4.1 Baselines and Benchmarks

**Benchmarks and metrics.** We trained our models on the BIRD training set (Li et al., 2024c), comprising 9,428 Question-SQL pairs from 70 databases across domains like airlines, movies, and sales. To address known issues with noisy and ambiguous queries (Pourreza et al., 2024; Talaei et al., 2024; Li et al., 2024b), we filtered out samples flagged as incorrect by both Gemini-2.0-flash and GPT-4o, resulting in 8,026 training examples. For evaluation, we primarily use the BIRD benchmark and include Spider, Spider-DK (Gan et al., 2021b), and Spider-Syn (Gan et al., 2021a) to assess generalization. While BIRD captures real-world SQL challenges with diverse schemas and noisy queries, Spider provides an out-of-distribution test with queries from a broad range of databases. Spider-Syn tests robustness through paraphrased questions using schema-related synonyms. Spider-DK modifies Spider queries with added domain knowledge to reflect realistic paraphrasing. We report execution accuracy (EX), requiring exact row match.

**Training settings.** We employ open-source models from the Qwen2.5-Coder (Hui et al., 2024) family, specifically the 3B, 7B, and 14B variants, for our experiments. We select this model family primarily due to its strong performance and the availability of multiple model sizes, allowing us to adapt to different computational constraints and use cases. In our work, a model trained with all of our proposed partial rewards is denoted with the subscript ($allrewards$) (e.g., Qwen2.5-coder-7B$_{(\text{all rewards})}$). Since most small LLMs, such as those used in this paper, have limited context window sizes—and to decouple schema linking from SQL generation for improved performance (Li et al., 2024b; Pourreza & Rafiei, 2024b)—we first filter the database schema using Gemini-1.5-pro for schema linking, following the method proposed in (Talaei et al., 2024). To ensure that the filtered schema contains all necessary information, we combine it with the ground truth schema used in the corresponding SQL query. However, during inference, as the ground truth answer is unavailable, we only filter the database schema using Gemini-1.5-pro to provide the model with a smaller, more manageable schema. he weights assigned to the partial rewards in this work are as follows: $w_{\text{exec}} = 3$, $w_{\text{judge}} = 2$, and all other partial reward weights are set to 1. More details of the training hyperparameters and settings are provided in Appendix section A.1.

**Text-to-SQL pipelines.** State-of-the-art Text-to-SQL methods typically use multi-stage pipelines rather than a single LLM call, incorporating steps like value retrieval, schema linking, query generation, self-correction, and selection (Pourreza et al., 2024; Gao et al., 2024; Lee et al., 2024; Wang et al., 2023). For fair comparison, we integrate our trained models into the CHASE-SQL pipeline (Pourreza et al., 2024), replacing Gemini-1.5-Pro. Following CHASE-SQL's setup, we train a binary pairwise verifier utilizing the open-source Qwen2.5-Coder 14B for query selection.

## 5 Results

### 5.1 Training Objective Ablation

In this section, we provide a detailed analysis of the performance of the Qwen2.5-Coder-7B model trained with different reward functions and compare it against models trained using SFT and the base Qwen2.5-Coder-Instruct model (prompts for training are provided in Appendix A). For SFT, we consider direct SQL prediction as well as the STaR-based (Zelikman et al., 2022) approach, which includes a CoT reasoning step before generating the SQL query. In our GRPO experiments, we explore various combinations of the reward functions described in Section 3.2. Additionally, to evaluate the importance of step-by-step reasoning for SQL generation, we report results from GRPO training on a model that was not required to output these reasoning steps. To ensure consistency in evaluation, we use Gemini-1.5-Pro-002 for schema linking, following the CHESS framework (Talaei et al., 2024), and filter the database schema before feeding it into the models. We further report the filtered-schema performance by providing the correct database schema for each question, representing the best achievable performance given a perfect schema linking module.

Table 1: Execution Accuracy (EX) of the Qwen2.5-Coder-7B model trained with different RL rewards and SFT datasets. In addition to overall EX and filtered-schema (FS) EX, we report the Syntax Check, Jaccard Similarity of Schema Items, and Jaccard Similarity of N-gram metrics. The upward arrows and delta values indicate the direct improvement in the corresponding metric when its associated reward is added.

| Post-training Method | Model | Thinking | Syntax | Schema | N-gram | EX (%) | FS EX (%) |
|---|---|---|---|---|---|---|---|
| No Post-training | Qwen2.5-Coder-7B | ✗ | 93.02 | 91.57 | 56.09 | 58.73 | 64.01 |
| SFT | Qwen2.5-coder-7B SFT | ✗ | 96.80 | 90.20 | 59.54 | 61.53 | 69.23 |
| | Qwen2.5-coder-7B STaR-SFT | ✓ | 94.71 | 90.71 | 58.02 | 62.84 | 68.12 |
| GRPO training | Qwen2.5-coder-7B-no-reasoning$_{(\textbf{all rewards})}$ | ✗ | 94.71 | 92.03 | 58.02 | 62.25 | 68.12 |
| | Qwen2.5-coder-7B$_{(exe)}$ | ✓ | 95.04 | 92.17 | 57.62 | 62.32 | 68.90 |
| | Qwen2.5-coder-7B$_{(exe,\textbf{syn})}$ | ✓ | **96.41** | 92.17 | 58.34 | 62.84 | 68.97 |
| | Qwen2.5-coder-7B$_{(exe,\textbf{schema})}$ | ✓ | 95.34 | **93.04** | 58.12 | 63.10 | 69.03 |
| | Qwen2.5-coder-7B$_{(exe,\textbf{n-gram})}$ | ✓ | 95.78 | 92.26 | 62.08 | 63.03 | 68.9 |
| | Qwen2.5-coder-7B$_{(exe,syn,\textbf{schema})}$ | ✓ | 95.76 | 92.95 | 58.34 | 63.10 | 69.23 |
| | Qwen2.5-coder-7B$_{(exe,syn,schema,\textbf{ngram})}$ | ✓ | 96.80 | 92.85 | **62.26** | 63.75 | 69.94 |
| | Qwen2.5-coder-7B$_{(Only\ RLAIF)}$ | ✓ | 95.56 | 92.26 | 58.10 | 63.75 | 69.16 |
| | Qwen2.5-coder-7B$_{(\textbf{all rewards})}$ | ✓ | 96.21 | 92.91 | 61.18 | **64.01** | **70.66** |

As shown in Table 1, using our proposed partial reward functions yields higher performance compared to training with the sparse execution accuracy reward alone. Notably, the addition of each reward directly improves its corresponding metric. For example, adding the syntax reward boosts the Syntax Check score by 1.37, incorporating the schema reward raises the Jaccard Similarity of Schema Items by 0.78, and including the N-gram reward increases the Jaccard Similarity of N-gram by 3.92. These improvements, highlighted with upward arrows and delta values, emphasize that each reward has a targeted effect while simultaneously contributing to the overall execution accuracy. Overall, our proposed method improves the base model's performance by 6.77%, whereas conventional SFT training alone results in a 4.11% performance gain. Moreover, comparing our GRPO-trained model with reasoning against the model that is GRPO-trained but only predicts the SQL query, we observe a 2% performance gain, demonstrating the importance of step-by-step reasoning for SQL generation. We further investigate the behavior of models under varying test-time compute budgets, with additional details in Appendix A.4.

## 5.2 Baselines Comparison

**Models comparison.** We report the performance of Qwen2.5-Coder models with 3B, 7B, and 14B parameters, trained using GRPO with all partial rewards on the BIRD development set. We compare these models against their respective base versions trained with SFT, as well as the base model itself. Additionally, we benchmark their performance against SOTA models, including o3-mini, and Gemini-2.0-Flash. Following the approach in Section 5.1, we evaluate execution accuracy with schema linking, where Gemini-1.5-Pro-002 serves as the schema linker together with filtered-schema accuracy, which is measured using the correct schema. As shown in Table 2, models trained with our proposed reward functions consistently outperform their SFT-trained counterparts, with a gap scaling with the model size (indicated by up arrows). More importantly, the 14B model surpasses the latest reasoning-focused model, o3-mini, by a significant margin of 4%. The only model that performs on par with our largest model is Gemini-2.0-Flash. This demonstrates the effectiveness in introducing reasoning capabilities, achieving SOTA performance with a 14B model that can be deployed on a single GPU.

Table 2: Execution Accuracy (EX) of various models trained with all RL rewards, the SFT model, and selected SOTA models. EX is reported with schema linking, while the filtered-schema EX is measured using the correct schema.

| Model | Thinking | EX (%) | Filtered-Schema EX (%) |
|---|---|---|---|
| GPT-4o-mini | ✗ | 60.82 | 66.29 |
| o3-mini | ✓ | 61.34 | 69.88 |
| Gemini-2.0-flash | ✗ | **66.49** | 71.9 |
| Gemini-1.5-pro-002 | ✗ | 62.25 | 67.47 |
| Qwen2.5-coder-3B | ✗ | 45.17 | 48.17 |
| Qwen2.5-coder-3B SFT | ✗ | 55.9 | 63.62 |
| Qwen2.5-coder-3B$_{(\textbf{all rewards})}$ | ✓ | 58.67 | 65.31 ↑ +1.69 |
| Qwen2.5-coder-7B | ✗ | 58.73 | 64.01 |
| Qwen2.5-coder-7B SFT | ✗ | 61.53 | 68.12 |
| Qwen2.5-coder-7B$_{(\textbf{all rewards})}$ | ✓ | 64.01 | 70.66 ↑ +2.54 |
| Qwen2.5-coder-14B | ✗ | 63.1 | 68.77 |
| Qwen2.5-coder-14B SFT | ✗ | 63.75 | 68.83 |
| Qwen2.5-coder-14B$_{(\textbf{all rewards})}$ | ✓ | **65.31** | **72.03** ↑ +3.20 |

**Reasoning-SQL vs. previous text-to-SQL methods.** Recent Text-to-SQL systems, particularly on benchmarks like BIRD, adopt pipeline frameworks where SQL generation is just one part of a larger process (Pourreza et al., 2024; Gao et al., 2024; Xie et al., 2025; Lee et al., 2024; Wang et al., 2023). These pipelines typically incorporate schema linking (to simplify schemas and reduce LLM confusion (Talaei et al., 2024)), initial SQL generation using filtered schemas, iterative self-correction, and a query selection mechanism—either model-based (Pourreza et al., 2024; Gao et al., 2024) or self-consistency driven (Sun et al., 2023a; Gao et al., 2023). To assess our GRPO-trained models in this context, we integrate them into the CHASE-SQL framework for schema linking, SQL generation, and self-correction, and replace Gemini-1.5-pro with a fine-tuned Qwen2.5-Coder-14B model for query selection. As shown in Table 3, our approach achieves 72.29% and 72.78% execution accuracy on the BIRD development and test sets, respectively. Our approach bridges the gap between open-source and proprietary solutions, positioning it as the state-of-the-art among small open-source models while outperforming many GPT-based systems. Additionally, it is important to note that our approach attains performance on par with CHASE-SQL at 93% lower cost, as detailed in Appendix Section A.5.1.

Table 3: Execution Accuracy (EX) of our model with the CHASE-SQL pipeline and comparison with all methods on the BIRD development set. "All" denotes the use of all introduced rewards combined.

| Method | Model | Model Size | Dev EX (%) | Test EX (%) |
|---|---|---|---|---|
| CHASE-SQL (Pourreza et al., 2024) | Gemini-1.5-pro | Unknown | 74.46 | 74.79 |
| XiYan-SQL (Gao et al., 2024) | Ensemble of Models | Unknown | 73.34 | 75.63 |
| OpenSearch-SQL v2 (Xie et al., 2025) | GPT-4o | Unknown | 69.3 | 72.28 |
| CHESS$_{IR+CG+UT}$ (Talaei et al., 2024) | Gemini-1.5-pro | Unknown | 68.31 | 71.10 |
| Distillery (Maamari et al., 2024) | GPT-4o | Unknown | 67.21 | 71.83 |
| XiYan-SQL (Gao et al., 2024) | QwenCoder-32B | 32B | 67.01 | 69.03 |
| E-SQL (Caferoğlu & Ulusoy, 2024) | GPT-4o | Unknown | 65.68 | 66.29 |
| CodeS-15B (Li et al., 2024b) | CodeS | 15B | 58.47 | 60.37 |
| Think2SQL (Papicchio et al., 2025) | Qwen2.5-7B | 7B | 58.1 | – |
| DTS-SQL (Pourreza & Rafiei, 2024b) | DeepSeek-7B | 7B | 55.8 | 60.31 |
| Reasoning-SQL (Ours) | Qwen2.5-14B$_{(\textbf{all rewards})}$ | 14B | **72.29** | **72.78** |
| Reasoning-SQL (Ours) | Qwen2.5-7B$_{(\textbf{all rewards})}$ | 7B | **68.05** | – |
| Reasoning-SQL (Ours) | Qwen2.5-coder-7B STaR-SFT | 7B | 66.92 | – |
| Reasoning-SQL (Ours) | Qwen2.5-coder-7B | 7B | 66.12 | – |

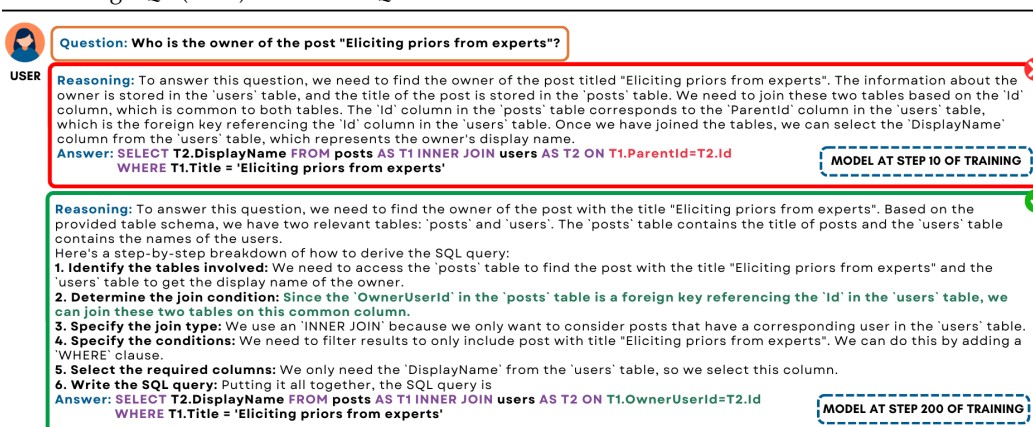

Figure 3: An example illustrating the model's improvement in reasoning. By step 200 of training, the model adopts a structured approach to SQL synthesis and correctly identifies the join condition.

## 5.3 Emergent Reasoning

In Section 5.1, we demonstrate that our reasoning model Qwen2.5-coder-7B$_{(\textbf{all rewards})}$ outperforms other baselines. Here, we further examine the reasoning strategy the model has developed. As illustrated in Figure 3, during early training stages the base model produces a non-structured, sub-optimal reasoning trace for synthesizing the SQL query, resulting in an incorrect "JOIN" operation. However, as training progresses, the model adopts a more structured reasoning style, yielding the correct SQL output. Notably, we do not enforce any particular reasoning format – rather, the effective reasoning style *emerges* naturally as the model optimizes for our designed rewards (for more examples see Appendix A.2). For further insights into the evolution of execution accuracy and reasoning efficiency during training, please refer to Appendix A.6.

We then ask whether the emergent reasoning pattern outperforms human-designed thinking formats. We generate thinking traces using three human-designed strategies, Divide-and-

Conquer (DC), Query Plan (QP) (Pourreza et al., 2024), and ACT-SQL (Zhang et al., 2023), for three randomly selected instances from the BIRD training set. Then, for each set of traces, the three human-designed strategies and our model's emergent reasoning, we design corresponding prompts using these thinking traces as few-shot examples. Finally, we evaluate the in-context learning performance of two base models, Qwen-14B-code-instruct and CodeGemma-7b-it, on the BIRD dev set. As shown in Table 4, prompts based on the emergent Reasoning-SQL style consistently outperform those using human-designed strategies for the two base models, highlighting both the efficacy of the learned reasoning style and its transferability across different models.

Table 4: Performance comparison generated from emergent reasoning vs. human-designed strategies on the BIRD dev set.

| Model | Prompt | Dev EX(%) |
|---|---|---|
| Qwen-14B code-inst | ACT-SQL | 62.84 |
| | Divide-and-Conquer | 60.75 |
| | Query Plan | 61.92 |
| | Reasoning-SQL | 64.21 |
| CodeGemma 7B-it | ACT-SQL | 48.37 |
| | Divide-and-Conquer | 48.82 |
| | Query Plan | 48.56 |
| | Reasoning-SQL | 50.19 |

Table 5: Comparison of performance on Spider, Spider-DK, and Spider-Syn (EX).

| Model | Thinking | Spider EX (%) | Spider-DK EX (%) | Spider-Syn EX (%) |
|---|---|---|---|---|
| GPT-4o-mini | ✗ | 76.3 | 71.02 | 67.02 |
| o3-mini | ✓ | 78.82 | 71.77 | 73.01 |
| Gemini-2.0-flash | ✗ | 81.23 | 74.15 | 73.59 |
| Gemini-1.5-pro-002 | ✗ | 80.46 | 67.66 | 71.56 |
| Qwen2.5-coder-7B | ✗ | 77.85 | 66.54 | 66.24 |
| Qwen2.5-coder-7B SFT | ✗ | 68.08 | 62.8 | 51.93 |
| Qwen2.5-coder-7B$_{(all rewards)}$ | ✓ | 78.72 | 73.27 | 69.34 |
| Qwen2.5-coder-14B | ✗ | 79.40 | 71.4 | 72.14 |
| Qwen2.5-coder-14B SFT | ✗ | 75.04 | 69.15 | 61.02 |
| Qwen2.5-coder-14B$_{(all rewards)}$ | ✓ | **81.43** | 73.03 | 72.63 |

## 5.4 Generalization Analysis

We evaluate the generalization capabilities of our approach on three benchmarks, Spider, Spider-DK, and Spider-Syn, which feature distinct SQL query and natural language question distributions. To ensure a fair comparison, we first performed schema linking using the Gemini-1.5-Pro model to isolate the relevant schema components, as detailed in Section 5.1. As shown in Table 5, our GRPO-trained model outperforms state-of-the-art alternatives like o3-mini and Gemini-1.5-Pro with on all three benchmarks. Furthermore, our experiments reveal that while our RL-trained models consistently improve upon the base models, the SFT-trained models underperform even relative to the base, confirming our observation that RL fosters robust generalization whereas SFT tends to favor memorization (Chu et al., 2025).

## 6 Conclusion

In this paper, we introduce a novel RL-based approach for the Text-to-SQL task, focusing on addressing critical reasoning subtasks, such as schema comprehension, query generation, and self-correction. Leveraging a set of carefully designed partial rewards and employing GRPO, we train different LLMs, significantly improving their reasoning and generalization capabilities. Our extensive experiments on benchmarks such as BIRD, Spider, Spider-DK, and Spider-SYN demonstrate that our RL-trained models consistently outperform models trained through SFT. Notably, our 14B-parameter model achieves state-of-the-art performance on the BIRD benchmark, surpassing significantly larger proprietary models. Our work underscores the potential of RL-based training methods and partial rewards to significantly advance the performance and generalization of smaller, open-source LLMs, reducing the performance gap between proprietary and open-source solutions in Text-to-SQL applications and enabling reasoning abilities.

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

# A  Appendix

## A.1  Training details

In this section, we provide detailed information about the training hyperparameters and settings utilized in this study. For training the Qwen-2.5-Coder models with GRPO, we employed a small learning rate of 1e-6 using a constant learning rate scheduler with a 0.1% warm-up ratio, an effective batch size of 32, and trained the models for 3 epochs on the filtered training dataset. For each input prompt, we generated 6 completions as the group size requirement for GRPO training. All GRPO training experiments were conducted on 8 NVIDIA H100 GPUs. We observed during GRPO training that formatting-related rewards were optimized first, followed by the task-specific rewards, highlighting an interesting learning pattern within the model.

For supervised fine-tuning (SFT), we used a comparatively larger learning rate of 1e-5 with a cosine learning rate scheduler, maintaining the same effective batch size of 32 and training for 3 epochs.

## A.2  Example Output

In this section, we provide an example output from the Qwen2.5-Coder-14B$_{(\text{all rewards})}$ model, demonstrating the step-by-step reasoning performed by the model prior to generating the final SQL query. This reasoning varies for each question and adapts according to the conditions required by the model.

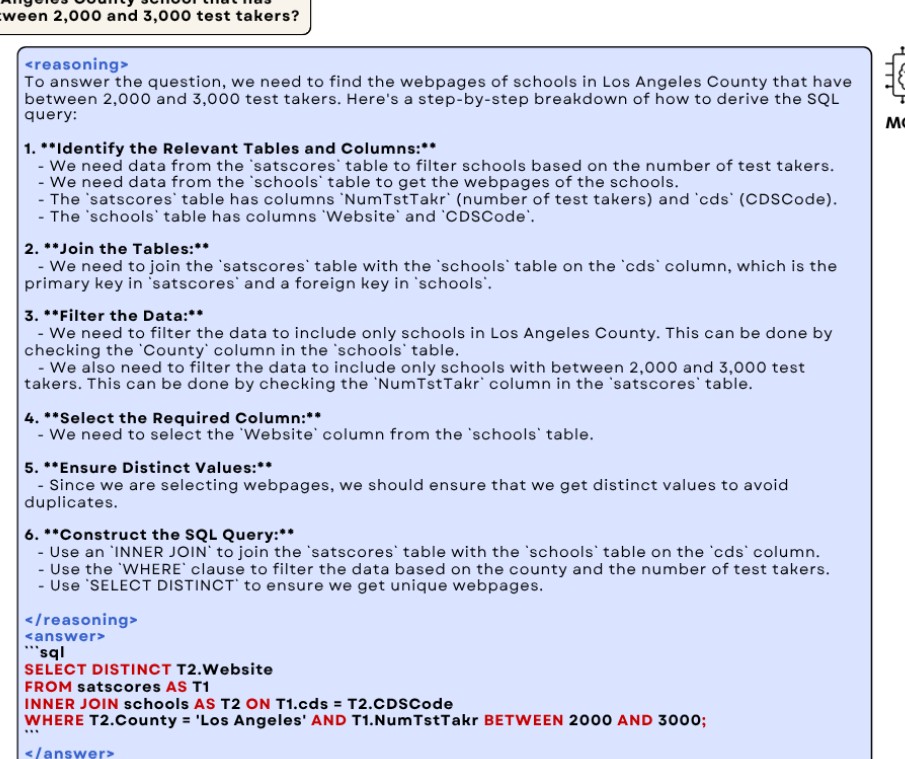

Figure 4: An example output of our GRPO-trained model.

Below we also provide another example output of the model during the early steps of the training (step 10) and compare it with the reasoning output of the model at one epoch of training.

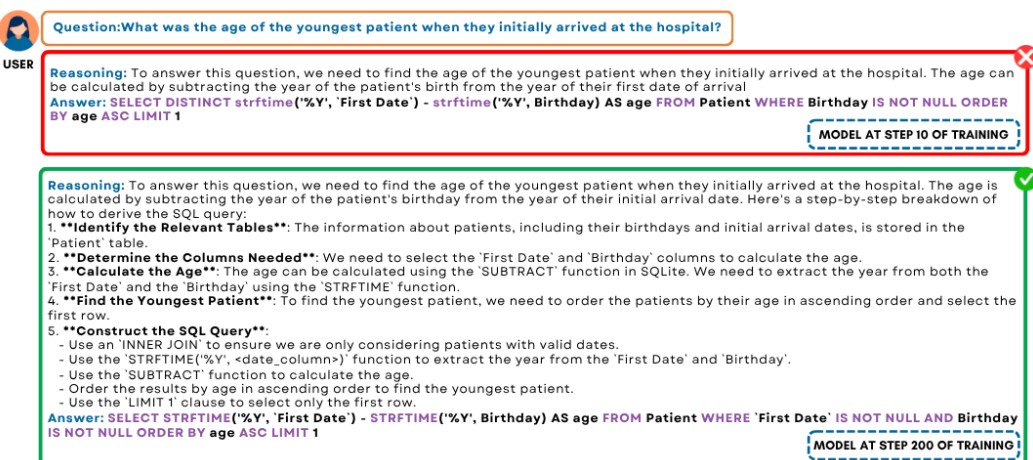

Figure 5: An example output of the model during the training process at step 10 and step 200.

### A.2.1 Model Checkpoints

In this section, we present the outputs generated by the Qwen2.5-Coder-3B$_{(\textbf{all rewards})}$ model at checkpoints 250, 500, 750, and the final checkpoint at 849, where the output at the last checkpoint provides the correct answer. The outputs correspond to the following question:

Question: In race No. 45, what is the abbreviated code of the driver whose Q3 time was 0:01:33?

**Checkpoint-250-output**

**<reasoning>**
To answer this question, we need to find the abbreviated code for the driver who had a specific Q3 time for race No. 45. The key columns involved are 'raceId' and 'driverId', which are foreign keys in the 'qualifying' table that reference the 'driverId' in the 'drivers' table. We will use a 'JOIN' clause to link these tables. The 'raceId' is specified as 45 in the query, and the 'q3' column is filtered to only include times with the format 'M:SS%', which indicates the Q3 time.
Here's a step-by-step breakdown of how to derive the SQL query:

1. **Identify the relevant tables**: We need to access data from the 'qualifying' table to get the Q3 time and the 'drivers' table to get the abbreviated code.
2. **Determine the join condition**: The 'qualifyId' in the 'qualifying' table is the same as the 'driverId' in the 'drivers' table, so we will join these two tables on this common column.
3. **Specify the join type**: Since we want to find a specific driver, we will use an 'INNER JOIN' to ensure that we only consider rows where there is a matching driver.
4. **Filter by race number**: We will filter the results to only include race No. 45 by adding a 'WHERE' clause with 'raceId = 45'.
5. **Filter by Q3 time**: We will filter the results to only include drivers who have a Q3 time of 'M:SS%', which indicates the Q3 time.
6. **Select the required columns**: We will select the 'code' column from the 'drivers' table, which contains the abbreviated code for the driver.
7. **Use 'DISTINCT'**: Since we are interested in unique drivers, we will use the 'DISTINCT' keyword to ensure that each driver's abbreviated code is listed only once.

**</reasoning>**
**<answer>**
"'sql
SELECT DISTINCT T2.code FROM qualifying AS T1 INNER JOIN drivers AS T2 ON T1.driverId = T2.driverId WHERE T1.raceId = 45 AND T1.q3 LIKE 'M:SS"'
**</answer>**

**Checkpoint-500-output**

**<reasoning>**
To answer this question, we need to find the abbreviated code for the driver who participated in race No. 45 and had a Q3 time of 0:01:33. Here's a step-by-step breakdown of how we derive the SQL query:
1. **Identify the relevant tables**: We need to access the 'drivers' table to get the abbreviated codes and the 'qualifying' table to get the Q3 times for each race.
2. **Determine the join condition**: The common column between the two tables is 'driverId', which links each driver to their qualifying times.
3. **Specify the join type**: Since we need to link each driver to their qualifying times, we use an 'INNER JOIN' between the 'drivers' and 'qualifying' tables on the 'driverId' column.
4. **Filter the race**: We need to filter the results to only include the race with a raceId of 45. This is done using a 'WHERE' clause on the 'raceId' column.
5. **Filter the driver**: We need to filter the results to only include the driver who had a Q3 time of 0:01:33. This is done using a 'WHERE' clause on the 'q3' column, which matches the pattern 'M:SS%'.
6. **Select the abbreviated code**: The abbreviated code is stored in the 'code' column of the 'drivers' table. We select this column from the 'drivers' table.
7. **Construct the query**: Combining all the steps, we get the SQL query as follows
**</reasoning>**
**<answer>**
"'sql
SELECT T1.code FROM drivers AS T1 INNER JOIN qualifying AS T2 ON T1.driverId = T2.driverId WHERE T2.raceId = 45 AND T2.q3 = '1:01:33'
"'
**</answer>**

**Checkpoint-750-output**

**<reasoning>**
To answer this question, we need to find the abbreviated code of the driver who had a specific Q3 time in race No. 45. Here's a step-by-step breakdown of how we derive the SQL query:
1. **Identify the relevant tables**: We need information from both the 'drivers' and 'qualifying' tables. The 'drivers' table contains the abbreviated code of the drivers, and the 'qualifying' table contains the Q3 times for each driver and race.
2. **Determine the join condition**: The 'driverId' in the 'drivers' table is the foreign key in the 'qualifying' table, which links the two tables. We will use this as the join condition to link the driver's code with their Q3 times.
3. **Specify the join type**: We will use an 'INNER JOIN' to combine the rows from both tables where the join condition is met.
4. **Filter the results**: We need to filter the results to only include the race No. 45. This can be done by adding a 'WHERE' clause to the query with the condition 'raceId = 45'.
5. **Filter the results further**: We need to filter the results to only include the driver who had a specific Q3 time. This can be done by adding a 'WHERE' clause to the query with the condition 'q3 LIKE '0:01:33%''.
6. **Select the required columns**: We need to select the abbreviated code of the driver. This can be done by adding a 'SELECT' statement to the query with the column 'code' from the 'drivers' table.
7. **Construct the SQL query**: Based on the above steps, the SQL query can be constructed as follows
**</reasoning>**
**<answer>**
"'sql
SELECT T1.code FROM drivers AS T1 INNER JOIN qualifying AS T2 ON T1.driverId = T2.driverId WHERE T2.raceId = 45 AND T2.q3 LIKE '0:01:33%'
"'

**</answer>**

**Checkpoint-849-output**

**<reasoning>**
To answer this question, we need to find the abbreviated code of a specific driver in a given race where the Q3 time is 0:01:33. Let's break down the steps required to derive the SQL query:
1. **Identify the relevant tables**: We need to access the 'drivers' table for the abbreviated code and the 'qualifying' table to get the Q3 time.
2. **Determine the join condition**: The 'driverId' in the 'qualifying' table is the primary key that links it to the 'driverId' in the 'drivers' table. Therefore, we will join the 'qualifying' table with the 'drivers' table on this common column.
3. **Specify the join type**: Since we need to find the driver with a specific Q3 time, we will use an 'INNER JOIN' to ensure that we only consider drivers who have a matching record in the 'qualifying' table.
4. **Filter by raceId**: We need to filter the results to only include the race with ID 45. This is done using the 'raceId' column in the 'qualifying' table.
5. **Filter by Q3 time**: We need to filter the results to only include the driver with a Q3 time of 0:01:33. This is done using the 'q3' column in the 'qualifying' table, with the condition 'q3 LIKE '1:33%''.
6. **Select the abbreviated code**: We need to select the 'code' column from the 'drivers' table, which contains the abbreviated code of the drivers.
Therefore, the final SQL query is
**</reasoning>**
**<answer>**
"'sql
SELECT T2.code FROM qualifying AS T1 INNER JOIN drivers AS T2 ON T1.driverId = T2.driverId WHERE T1.raceId = 45 AND T1.q3 LIKE '1:33%'
"'

**</answer>**

## A.3 Prompts

---

**LLM-as-a-Judge Prompt**

*You are an experienced database expert. Your task is to evaluate a generated SQL query by comparing it to the ground truth (gold) query and then assign a score between 0.0 and 2.0. A higher score indicates the predicted query is more correct, while a score of 0.0 means it is completely incorrect.*

**Follow these evaluation rules strictly:**

1. **SELECT Clause:**
   - Only select columns that are mentioned in the user's question.
   - Do not include unnecessary columns or values.

2. **Aggregation (MAX/MIN):**
   - Always perform JOINs before applying MAX() or MIN().

3. **ORDER BY with Distinct Values:**
   - Use a `GROUP BY <column>` before an `ORDER BY <column> ASC|DESC` to ensure distinct values.

4. **Handling NULLs:**
   - If a column may contain NULL values (indicated by "None" in value examples or explicitly mentioned), include a `JOIN` or a `WHERE <column> IS NOT NULL` clause.

5. **FROM/JOIN Clauses:**
   - Only include the tables essential for answering the question.

6. **Strictly Follow Hints:**
   - Adhere to all hints provided with the question.

7. **Thorough Question Analysis:**
   - Ensure all conditions and requirements mentioned in the question are addressed.

8. **DISTINCT Keyword:**
   - Use `SELECT DISTINCT` when the question requires unique values (e.g., IDs, URLs) or when column statistics (Value Statics) indicate its necessity.

9. **Column Selection:**
   - Carefully analyze column descriptions and hints to choose the correct column when similar columns exist across tables.

10. **String Concatenation:**
    - Do not use any string concatenation methods (e.g., `|| ' ' ||`) in the `SELECT` clause.

11. **JOIN Preference:**
    - Prefer using `INNER JOIN` over nested `SELECT` statements.

12. **Date Processing:**
    - Use `STRFTIME()` for any date manipulations (e.g., `STRFTIME('%Y', SOMETIME)` to extract the year).

**You are provided with the following inputs:**

- **Question:** {QUESTION}
- **Hint:** {HINT}
- **Gold Query:** {GOLD_QUERY}
- **Predicted Query:** {PREDICTED_QUERY}

**Based on the above, return a single numeric score between 0.0 and 2.0 that reflects how correct the predicted query is compared to the gold query. Respond with only the score and no additional explanation.**

---

**Database Admin Instructions**

1. **SELECT Clause:**
   - Only select columns mentioned in the user's question.
   - Avoid unnecessary columns or values.

2. **Aggregation (MAX/MIN):**
   - Always perform JOINs before using MAX() or MIN().

3. **ORDER BY with Distinct Values:**
   - Use `GROUP BY <column>` before `ORDER BY <column> ASC|DESC` to ensure distinct values.

4. **Handling NULLs:**
   - If a column may contain NULL values (indicated by "None" or explicitly mentioned), include a `JOIN` or `WHERE <column> IS NOT NULL`.

5. **FROM/JOIN Clauses:**
   - Include only essential tables for answering the question.

6. **Strictly Follow Hints:**
   - Adhere to all provided hints.

7. **Thorough Question Analysis:**
   - Address all conditions mentioned in the question.

8. **DISTINCT Keyword:**
   - Use `SELECT DISTINCT` when unique values (e.g., IDs, URLs) are needed.

9. **Column Selection:**
   - Analyze column descriptions and hints carefully to choose correctly when similar columns exist.

10. **String Concatenation:**
    - Never use `|| ' ' ||` or other concatenation in `SELECT`.

11. **JOIN Preference:**
    - Prioritize `INNER JOIN` over nested `SELECT` statements.

12. **Date Processing:**
    - Use `STRFTIME()` for date manipulations (e.g., `STRFTIME('%Y', SOMETIME)`).

**SQL Generation Divide-and-Conquer Prompt (USED FOR STaR-SFT)**

*You are an experienced database expert. Now you need to generate a SQLite query given the database information, a question and some additional information. The database structure is defined by table schemas (some columns provide additional column descriptions in the options).*

Given the table schema information description and the `Question`, you will be given table creation statements and you need to understand the database and columns.

You will be using a method called "recursive divide-and-conquer approach to SQL query generation from natural language."

**Here is a high-level description of the steps:**

1. **Divide (Decompose Sub-question with Pseudo SQL):** The complex natural language question is recursively broken down into simpler sub-questions. Each sub-question targets a specific piece of information or logic required for the final SQL query.

2. **Conquer (Real SQL for sub-questions):** For each sub-question (and the main question initially), a "pseudo-SQL" fragment is formulated. This pseudo-SQL represents the intended SQL logic but might have placeholders for answers to the decomposed sub-questions.

3. **Combine (Reassemble):** Once all sub-questions are resolved and their corresponding SQL fragments are generated, the process reverses. The SQL fragments are recursively combined by replacing the placeholders in the pseudo-SQL with the actual generated SQL from the lower levels.

4. **Final Output:** This bottom-up assembly culminates in the complete and correct SQL query that answers the original complex question.

**Database admin instructions:** *[DATABASE ADMIN INSTRUCTIONS PLACEHOLDER]*

**Now is the real question, following the instruction and examples, generate the SQLite Query with Recursive Divide-and-Conquer approach. Follow all steps from the strategy. When you get to the final query, output the query string ONLY in the format "`sql ... `". Make sure you only output one single query.**

---

**Table creation statements**

{DATABASE_SCHEMA}

---

**Question**

{QUESTION} {HINT}

---

**Answer**

Repeating the question and generating the SQL with Recursive Divide-and-Conquer.

## SQL Reasoning Prompt (USED FOR GRPO)

**Instructions:** *You are an experienced database expert. Now you need to generate a SQL query given the database information, a question and some additional information. The database structure is defined by the following table schemas (comments after '−' provide additional column descriptions).*

Note that the "Example Values" are actual values from the column. Some columns might contain the values that are directly related to the question. Use this information to justify which columns to use.

Given the table schema information description and the `Question`, you will be given table creation statements and you need to understand the database and columns to generate a single SQLite query that can answer the user's question.

**Database admin instructions:** *[DATABASE ADMIN INSTRUCTIONS PLACEHOLDER]*

---

**[Table creation statements]**

  {DATABASE_SCHEMA}

---

**Now is the real question, following the instruction and examples, generate the SQL query.**

---

**Question:**

  {QUESTION} **Hint:** {HINT}

---

Respond in the following format:

```
<reasoning>
Your detailed and step-by-step thinking path toward finding the correct SQL query
</reasoning>
<answer>
"'sql
Your predicted SQL query
"'
</answer>
```

**Now is your turn to respond in the above format.**

## SQL Generation Supervised Fine-tuning Prompt (USED FOR SFT)

**Instructions:** *You are an experienced database expert. Now you need to generate a SQL query given the database information, a question and some additional information. The database structure is defined by the following table schemas (comments after '−' provide additional column descriptions).*

Note that the "Example Values" are actual values from the column. Some columns might contain values directly related to the question. Use this information to justify which columns to use.

Given the table schema information description and the **Question**, you will be provided table creation statements. You need to understand the database and columns to generate a single SQLite query that answers the user's question.

**Database admin instructions:** *[DATABASE ADMIN INSTRUCTIONS PLACEHOLDER]*

---

**[Question]**

  {QUESTION} {HINT}

---

Respond strictly in the following format:

```
"'sql
Your predicted SQL query
"'
```

---

**In-context learning example for Reasoning-SQL format**

Table creation statements:

```
CREATE TABLE Product
(
        ProductID INTEGER primary key autoincrement,
        StandardCost REAL not null, -- Example Values: `(0.0,)`
        Name TEXT not null unique, -- Example Values: `('AWC Logo Cap',)`
);

CREATE TABLE ProductCostHistory
(
        foreign key (ProductID) references Product(ProductID),
        StandardCost REAL not null, -- Example Values: `(12.0278,)`
        ProductID INTEGER not null, -- Example Values: `(707,)`
        StartDate DATE not null, --
        primary key (ProductID, StartDate),
);
```

**Question:**

List the products whereby the standard cost is $80 more than previous standard cost in history. Hint: SUBTRACT(product.StandardCost, CostHistory.StandardCost)>80

**Reasoning:**

To answer the question, we need to find products where the current standard cost is $80 more than their previous standard cost in history. Here's a step-by-step breakdown of how to derive the SQL query:

1. **Identify Relevant Tables and Columns:** - We need data from the `Product` table to get the product names and their current standard costs. - We need data from the `ProductCostHistory` table to get the historical standard costs.

2. **Determine the Join Condition:** - Join the `Product` table with `ProductCostHistory` on the `ProductID` column.

3. **Filter Based on the Condition:** - Include only those products where the current standard cost is $80 more than the previous standard cost.

4. **Select the Required Columns:** - Select the `Name` column from `Product`.

5. **Ensure Distinct Values:** - Ensure distinct values through the join condition.

**Answer:**

"'*sql*
SELECT DISTINCT P.Name FROM Product P INNER JOIN ProductCostHistory CH ON P.ProductID = CH.ProductID WHERE P.StandardCost - CH.StandardCost > 80;
"'

---

**In-context learning example for Query-prompt format**

Table creation statements:
```
CREATE TABLE Product
(
        ProductID INTEGER primary key autoincrement,
        StandardCost REAL not null, -- Example Values: `(0.0,)`
        Name TEXT not null unique, -- Example Values: `('AWC Logo Cap',)`
);

CREATE TABLE ProductCostHistory
(
        foreign key (ProductID) references Product(ProductID),
        StandardCost REAL not null, -- Example Values: `(12.0278,)`
        ProductID INTEGER not null, -- Example Values: `(707,)`
        StartDate DATE not null, --
        primary key (ProductID, StartDate),
);
```

**Question:**

List the products whereby the standard cost is $80 more than previous standard cost in history. Hint: SUBTRACT(product.StandardCost, CostHistory.StandardCost)>80

**Query Plan Guided SQL Generation:**
**Step-by-Step Reasoning:**

1. **Initialization:**
   - Open tables: Product, ProductCostHistory.

2. **Columns Needed:**
   - Product: StandardCost (current), Name.
   - ProductCostHistory: StandardCost (historical), ProductID.

3. **Table JOIN Operation:**
   - INNER JOIN on `Product.ProductID = ProductCostHistory.ProductID`.

4. **Filter Condition:**
   - Select products where current `StandardCost` exceeds historical `StandardCost` by more than $80:

     ```
     Product.StandardCost - ProductCostHistory.StandardCost > 80
     ```

5. **Select Final Column:**
   - DISTINCT product names (`Product.Name`).

**Final Optimized SQL Query:**

```sql
SELECT DISTINCT P.Name FROM Product P INNER JOIN ProductCostHistory CH ON P.ProductID = CH.ProductID WHERE P.StandardCost - CH.StandardCost > 80;
```

**In-context learning example for Divide-and-conquer format**

Table creation statements:

```
CREATE TABLE Product
(
        ProductID INTEGER primary key autoincrement,
        StandardCost REAL not null, -- Example Values: `(0.0,)`
        Name TEXT not null unique, -- Example Values: `('AWC Logo Cap',)
);

CREATE TABLE ProductCostHistory
(
        foreign key (ProductID) references Product(ProductID),
        StandardCost REAL not null, -- Example Values: `(12.0278,)`
        ProductID INTEGER not null, -- Example Values: `(707,)`
        StartDate DATE not null, --
        primary key (ProductID, StartDate),
);
```

**Question:**

List the products whereby the standard cost is $80 more than previous standard cost in history. Hint: SUBTRACT(Product.StandardCost, CostHistory.StandardCost)>80

**Divide and Conquer:**

1. **Main Question Analysis:**
   - Identify products whose current standard cost is at least $80 greater than historical costs.
   - Join tables `Product` and `ProductCostHistory` on `ProductID`.
   - Pseudo SQL: SELECT DISTINCT Product.Name FROM Product INNER JOIN ProductCostHistory ON Product.ProductID = ProductCostHistory.ProductID WHERE <condition>;

2. **Sub-question (StandardCost difference):**
   - Condition: `Product.StandardCost - ProductCostHistory.StandardCost > 80`

**Assemble SQL:**

- SQL condition: `Product.StandardCost - ProductCostHistory.StandardCost > 80`
- Complete SQL query:

**Simplification and Optimization:**

- Query is straightforward and optimized.

**Final SQL Query:**

```sql
SELECT DISTINCT P.Name FROM Product P INNER JOIN ProductCostHistory CH ON P.ProductID = CH.ProductID WHERE P.StandardCost - CH.StandardCost > 80;
```

---

**In-context learning example for ACT-SQL format**

Table creation statements:

```
CREATE TABLE Product
(
        ProductID INTEGER primary key autoincrement,
        StandardCost REAL not null, -- Example Values: `(0.0,)`
        Name TEXT not null unique, -- Example Values: `('AWC Logo Cap',)`
);

CREATE TABLE ProductCostHistory
(
        foreign key (ProductID) references Product(ProductID),
        StandardCost REAL not null, -- Example Values: `(12.0278,)`,
        ProductID INTEGER not null, -- Example Values: `(707,)`,
        StartDate DATE not null, --
        primary key (ProductID, StartDate),
);
```

**Question:**

List the products whereby the standard cost is \$80 more than previous standard cost in history. Hint: `SUBTRACT(product.StandardCost, CostHistory.StandardCost)>80`

**Query Plan Guided SQL Generation:**
**Reasoning:**

- According to *"list the products"*, columns [`Product.Name`] may be used.

- According to *"standard cost is \$80 more than previous standard cost"*, columns [`Product.StandardCost, ProductCostHistory.StandardCost`] may be used.

- Tables [`Product, ProductCostHistory`] will be joined using [`ProductID`].

- Condition [`(Product.StandardCost - ProductCostHistory.StandardCost) > 80`] must be applied.

**Final Answer:**

```sql
SELECT DISTINCT P.Name FROM Product AS P INNER JOIN ProductCostHistory AS CH ON P.ProductID = CH.ProductID WHERE (P.StandardCost - CH.StandardCost) > 80;
```

## A.4    Test-time Compute Analysis

Scaling test-time compute (Snell et al., 2024) involves allocating additional computational resources during inference through methods such as Chain-of-Thought prompting (Wei et al., 2022), the Best-of-N approach (generating N candidate answers and selecting the best one either programmatically via test cases or using a verifier model) (Lightman et al., 2023; Pourreza et al., 2024; Gao et al., 2024; Lee et al., 2024), or by simply increasing the number of reasoning tokens (e.g., adding "wait" tokens at the reasoning step) (Muennighoff et al., 2025). Among these methods, the Best-of-N approach is widely adopted in the Text-to-SQL domain (Pourreza et al., 2024; Gao et al., 2024; Lee et al., 2024). Two critical metrics influence the effectiveness of the Best-of-N method: the Pass@K performance, indicating the upper bound achievable by selecting from multiple candidates, and the average quality of the candidate pool (Average@K). To compare the performance of our GRPO-trained models against the standard SFT-trained models, we evaluate the Pass@K and Average@K metrics for the Qwen2.5-Coder-14B SFT model and our GRPO-trained Qwen2.5-Coder-14B$_{(all)}$ model in Figure 6. To enhance candidate diversity, we increased the sampling temperature to 0.5, resulting in a slight performance drop for the GRPO-trained model (from 65.5% to 64.66%). However, the drop was more significant for the SFT-trained model (from 63.75% to 60.82%). Notably, although the SFT model demonstrates greater randomness, thus achieving a slightly higher Pass@K, our GRPO-trained model maintains significantly higher average performance across various candidate sizes (K). These results indicate that the GRPO-trained model generates candidates of consistently higher quality, albeit with slightly reduced diversity.

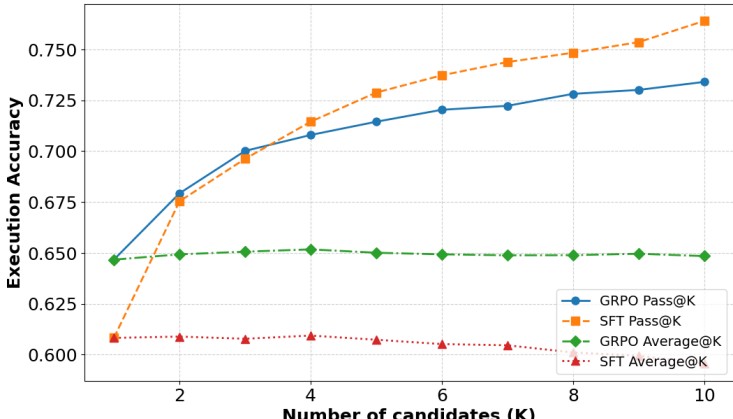

Figure 6: Pass@K and average@K performance for the GRPO and SFT trained models on bird development set.

## A.5    Difficulty and Cost Analysis

In this section, we report the performance of our GRPO-trained models with different sizes, 3B and 7B, across the three difficulty levels of questions in the BIRD development set and compare with STaR-SFT trained reasoning models. Additionally, we analyze the length of the reasoning portion in the LLM's output for each difficulty level. As shown in Table 6, performance of our GRPO-trained models are consistently better than STaR-SFT models across all difficulty levels with much less reasoning characters. Furthermore, for all models, the number of characters in the reasoning section of the output grows as the question difficulty increases, indicating a greater need for detailed reasoning in more complex queries. Furthermore, recognizing that the number of schema links required for a query serves as an indicator of query complexity (Pourreza & Rafiei, 2024a), we conducted an additional analysis comparing the length of reasoning characters generated by our models with the number of schema links. Our analysis revealed a positive correlation between these metrics, as illustrated in Figure 7.

Table 6: Execution Accuracy (EX) of our model with CHASE-SQL pipeline and comparison with all methods on BIRD development set. "All" denotes the use of all introduced rewards combined.

| Model | EX (%) | | | # Reasoning Chars | | |
|---|---|---|---|---|---|---|
| | Simple | Moderate | Challenging | Simple | Moderate | Challenging |
| Qwen2.5-coder-7B-STaR-SFT | 70.27 | 52.8 | 45.56 | 1550 | 2012 | 2346 |
| Qwen2.5-coder-7B$_{(all)}$ | 71.56 | 53.66 | 46.20 | 1180 | 1339 | 1421 |
| Qwen2.5-coder-3B-STaR-SFT | 62.48 | 43.1 | 38.62 | 1579 | 2030 | 2313 |
| Qwen2.5-coder-3B$_{(all)}$ | 67.78 | 46.76 | 38.62 | 1306 | 1606 | 1666 |

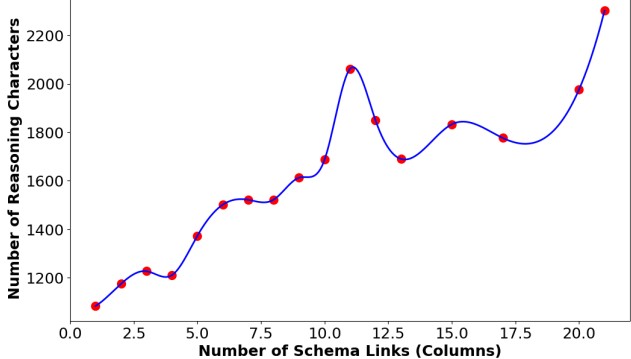

Figure 7: Length of reasoning segments generated by our GRPO-trained model across queries requiring varying numbers of schema links.

### A.5.1 Detailed Cost Analysis

In this section, we compare the cost of using our Qwen2.5-Coder-14B$_{(all)}$ model within the CHASE-SQL pipeline (Pourreza et al., 2024) against the original CHASE-SQL implementation, which utilizes Gemini-1.5-pro-002 for all LLM calls in its pipeline. Figure 8 illustrates the average cost comparison per question across each database. The results indicate that our model achieves comparable performance (1% lower than Gemini-1.5-pro) at a **93%** lower cost, highlighting the significant contribution of our work in reducing the expenses associated with proprietary models for SQL generation.For the Gemini-1.5-pro model, we estimated the token costs at approximately $1.25 per 1M input tokens and $5.00 per 1M output tokens. In contrast, our model's estimated costs are around $0.08 per 1M input tokens and $0.18 per 1M output tokens, contributing to this cost reduction.

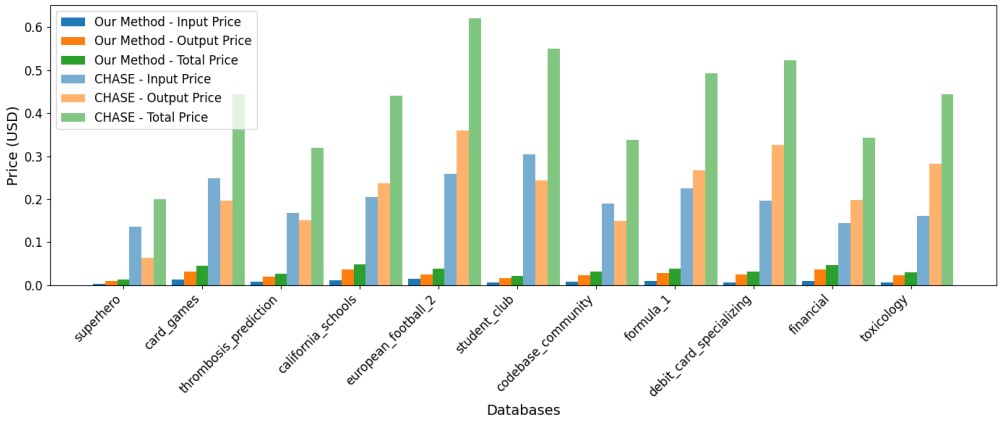

Figure 8: Cost analysis of using our Qwen2.5-coder-14B$_{(all)}$ vs using Gemini-1.5-pro-002 for the SQL generation with CHASE-SQL pipeline across different databases of the BIRD development set.

## A.6 Training Dynamics: Execution Accuracy and Reasoning Efficiency

In this section, we provide a detailed analysis of the model's execution accuracy and the corresponding number of reasoning characters generated during training. We conducted this analysis by evaluating the model's accuracy and reasoning length at intervals of 10 training steps, starting from 0 and continuing up to 200 steps. To perform an effective and efficient evaluation, we used a 10% sub-sample of the BIRD development set across these 20 evaluation points. Figure 9 illustrates that as training progresses, the accuracy of the model generally increases while the average number of reasoning characters tends to decrease. This reduction is partly due to our prompt's requirement for the model to always provide its reasoning steps before providing the answer. Initially, at early stages of training, the model leverages its zero-shot chain-of-thought capabilities to produce extensive reasoning steps. However, as training advances, the model increasingly standardizes and condenses its reasoning process, converging toward a more succinct and consistent format.

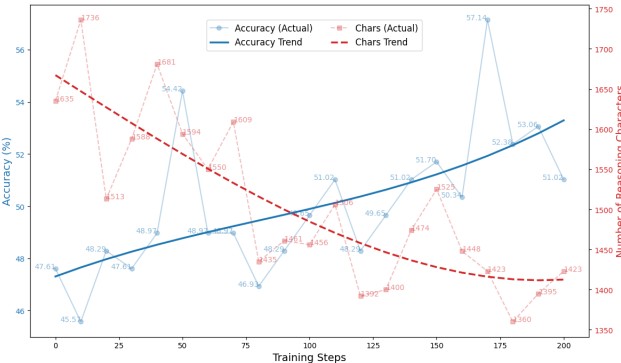

Figure 9: Accuracy and number of reasoning characters of the Qwen2.5-3B$_{(all)}$ model during the first 200 steps of the training.

## A.7 Pseudo-Code

In this section, we present the high-level pseudo-code detailing our GRPO-based reinforcement learning training algorithm 1 for Text-to-SQL. The pseudo-code outlines the generation of candidate SQL queries, the computation of a composite reward, including execution accuracy, LLM-based judge feedback with the judge prompt, schema linking, n-gram similarity, and format adherence, and the subsequent policy update via the GRPO objective. This detailed overview is provided to enhance reproducibility and to offer further clarity on the methodological aspects discussed in the main paper.

---

**Algorithm 1** GRPO-based RL Training for Text-to-SQL

---

**Input:**

- Policy model $\pi_\theta$ (initialized from a pretrained LLM)
- Reference model $\pi_{\text{ref}}$ (copy of the pretrained LLM)
- Old policy model $\pi_{\theta_{\text{old}}}$ (frozen snapshot of $\pi_\theta$ before update)
- Training dataset $\mathcal{D}$ consisting of question-schema pairs $(q, S)$ and gold SQL queries
- Number of candidates per example $G$
- Hyperparameters: clipping threshold $\epsilon$, divergence regularizer $\beta$
- Reward weights: $w_{\text{exec}}, w_{\text{judge}}, w_{\text{syntax}}, w_{\text{schema}}, w_{\text{ngram}}, w_{\text{format}}$

**Output:** Updated policy model $\pi_\theta$
**foreach** *training step* **do**
 **foreach** *mini-batch of examples* $(q, S, gold) \subset \mathcal{D}$ **do**
  **foreach** *example* $(q, S, gold)$ *in the mini-batch* **do**
   // Generate candidate SQL queries
   Sample $G$ candidates $\{o_i\}_{i=1}^G \sim \pi_\theta(o \mid q, S)$
   // Evaluate rewards for each candidate
   **foreach** *candidate* $o_i$ **do**
    $r_{\text{exec}} \leftarrow \mathbf{1}[\text{exec}(o_i) = \text{exec}(gold)]$
    **if** $r_{exec} = 0$ **then**
     $r_{\text{judge}} \leftarrow \text{LLM-judge}(o_i, gold) \in [0, 1]$
    **end**
    **else**
     $r_{\text{judge}} \leftarrow 1$
    **end**
    $r_{\text{syntax}} \leftarrow \mathbf{1}[\text{syntactically valid}(o_i)]$
    $r_{\text{schema}} \leftarrow \text{Jaccard}(\text{schema}(o_i), \text{schema}(gold))$
    $r_{\text{ngram}} \leftarrow \text{Jaccard}(\text{ngrams}(o_i), \text{ngrams}(gold))$
    $r_{\text{format}} \leftarrow \mathbf{1}[o_i \text{ matches required format}]$
    // Aggregate weighted rewards
    $r_i \leftarrow w_{\text{exec}} r_{\text{exec}} + w_{\text{judge}} r_{\text{judge}} + w_{\text{syntax}} r_{\text{syntax}}$
     $+ w_{\text{schema}} r_{\text{schema}} + w_{\text{ngram}} r_{\text{ngram}} + w_{\text{format}} r_{\text{format}}$
   **end**
   // Compute advantages
   **foreach** *candidate* $o_i$ **do**
    $A_i \leftarrow r_i - \frac{1}{G} \sum_{j=1}^G r_j$
   **end**
   // GRPO policy update
   Compute loss:

$$J_{\text{GRPO}}(\theta) \leftarrow \mathbb{E}\left[\frac{1}{G}\sum_{i=1}^G \min\left(\frac{\pi_\theta(o_i|q,S)}{\pi_{\theta_{\text{old}}}(o_i|q,S)} A_i, \text{clip}\left(\frac{\pi_\theta(o_i|q,S)}{\pi_{\theta_{\text{old}}}(o_i|q,S)}, 1-\epsilon, 1+\epsilon\right) A_i\right)\right]$$
$$- \beta D_{\text{KL}}(\pi_\theta \| \pi_{\text{ref}})$$

   Update $\theta \leftarrow \theta - \eta \nabla_\theta(-J_{\text{GRPO}}(\theta))$
   // Periodically update old policy snapshot
  **end**
 **end**
**end**

---

