# OpenReview forum: "Reasoning-SQL: Reinforcement Learning with SQL Tailored Partial Rewards for Reasoning-Enhanced Text-to-SQL"
_colmweb.org/COLM/2025/Conference — COLM 2025_

### Official Review · Reviewer_KUeS · 2025-04-19

**Rating:** 6
**Confidence:** 3
**Ethics Flag:** 1

**Summary:**

In this paper, the authors have proposed a training pipeline for reasoning-based text-to-SQL tasks. The authors have proposed several different rule-based rewards to improve performance of reinforcement learning. The experiments are extensive and show solid improvements compared with the baseline systems.

**Questions To Authors:**

When prompting LLMs, will the schema information of the underlying database also provided to the model? From the prompt examples, I didn't see such information.

**Reasons To Accept:**

1. The writing is easy to follow.
2. This study has covered some implementation details and experience that could be helpful to the community for further research, especially to the text-to-SQL research.
3. It validates the success and generalization of rule-based reinforcement learning.
4. [After Rebuttal] The updated experiments show improvements compared with llm-as-judge baseline built on the SFT policy.

**Reasons To Reject:**

1. Each single item of the rule-based rewards (i.e., syntax, schema linking, n-gram similarity, format) demonstrate marginal improvements. Considering the less stableness of reinforcement learning, I have some concern of whether these rewards can make some true contributions to the final performance.
2. Besides, by comparing the baselines using rule-based rewards with the one using only llm-as-judge, can we say that llm-as-judge can already bypass the hand-crafted reward design? Again, the improvements by applying rule-based rewards to llm-as-judge based model is somewhat marginal.

---

> ### Author Response · Authors · 2025-06-01
> **Authors' response**
>
> We would like to thank the reviewer for their constructive feedback. In response to your comments, we conducted the following experiments, which we hope adequately address your concerns regarding the paper:
>
> 1) We appreciate the reviewer’s thoughtful observation regarding the instability of RL training. To clarify, Table 1 in our paper reports results for five different runs with each of our partial rewards, each initialized from the same base policy. Each of our proposed partial rewards consistently led to performance improvements over the SFT baseline, indicating that the observed gains are robust and reproducible. Furthermore, to better demonstrate the effectiveness of our RL approach, we conducted an additional experiment where we applied our full reward design on top of the SFT policy model. This led to substantially higher performance, reinforcing the value of our composite reward strategy and its ability to amplify model performance beyond both the SFT and RL-only setups:
>
> | Model                                     | Ex    |
> |-------------------------------------------|-------|
> | Qwen2.5-coder-7B STaR-SFT (SFT only)      | 62.86 |
> | Qwen2.5-coder-7B (RL-only, all rewards)   | 64.01 |
> | Qwen2.5-coder-7B STaR-SFT (SFT + RL all reward)    | 65.5  |
>
> 2) To better demonstrate the performance gains achieved by combining rule-based rewards with the LLM-as-a-judge reward, we conducted an experiment comparing the Majority@k performance of a model trained with our full RL reward composition to that of a model trained using only the LLM-as-a-judge reward. The results show consistent improvements across all values of k with almost 2% gain, highlighting the importance and effectiveness of the composite reward design in enhancing overall model performance:
>
> https://anonymous.4open.science/r/REASONING-SQL-REBUTTAL-A664/majority@k.png
>
> Regarding your question about the database schema, we would like to clarify that the database schema is indeed an integral part of the prompt and is explicitly included within it. As mentioned with the following format in the prompts in appendix section:
> ```
> [Table creation statements]
>  {DATABASE_SCHEMA}
> ```

---

> > ### Comment · Reviewer_KUeS · 2025-06-09
> > **Thank you**
> >
> > Thanks for the efforts for addressing my concern. I will adjust my score accordingly.

---

> ### Author Response · Authors · 2025-06-07
> **Reminder**
>
> We’re writing to kindly follow up on our responses to your valuable comments. If you haven’t had a chance yet, we would greatly appreciate it if you could take a moment to review our responses and, if appropriate, update your scores accordingly.
>
> Thank you again for your time and valuable feedback throughout this process. Please don’t hesitate to let us know if any further clarification is needed from our side.

---

> > ### Comment · Area_Chair_PhCD · 2025-06-07
> > **Response to Author Rebuttal**
> >
> > Gentle reminder to please respond to the author's rebuttal

---

### Official Review · Reviewer_sLu2 · 2025-04-28

**Rating:** 7
**Confidence:** 3
**Ethics Flag:** 1

**Summary:**

The authors proposed a reinforcement learning (RL) based SQL generation system with large language model. The method utilizes a set of partial rewards, including, schema linking, AI feedback, n-gram and syntax check et al, to alleviate reward sparsity issues in RL.  Extensive experiments are conducted to evaluate the proposed method and it shows better results than the baselines and interesting emerging reasoning ability.

**Questions To Authors:**

Most partial rewards are based on gold SQL answers, which is also used in SFT baseline. Will the SFT baseline be helpful for the RL learning? For example, can SFT baseline is used as initialization or used as another reward?

Typo:
line 250 ``he weights --> The weights''

**Reasons To Accept:**

- The paper is well written and easy to follow in general.
- Extensive experiments have been conducted to demonstrate the ability of the proposed method, such as ablation study, different model sizes, and different evaluation settings etc.
- Good results are achieved

**Reasons To Reject:**

- Most of partial rewards, such as AI feedback, scheme linking, are n-gram similarity require gold SQL answers.  It limits the usage of the proposed method.

---

> ### Author Response · Authors · 2025-06-01
> **Authors' response**
>
> We would like to thank the reviewer for their constructive feedback.
>
> 1) We agree with the reviewer that access to gold SQL queries is required for computing the rewards. However, alternative approaches such as test case generation from natural language inputs—as proposed in [1]—or synthetic SQL generation, as demonstrated in Omni-SQL [2], can serve as auxiliary methods for SQL verification. These techniques offer promising directions for reducing dependency on gold queries during evaluation or training.
>
> To address your question regarding the effectiveness of using the SFT baseline as an initialization point, we conducted an additional experiment using Qwen 2.5 7B STaR SFT as the policy model to initialize training. The results, presented in the table below, show a clear performance gain on the BIRD dev set, demonstrating the effectiveness of our approach in boosting performance beyond both the SFT baseline and RL-only training:
>
> | Model                                     | Ex    |
> |-------------------------------------------|-------|
> | Qwen2.5-coder-7B STaR-SFT (SFT only)      | 62.86 |
> | Qwen2.5-coder-7B (RL-only, all rewards)   | 64.01 |
> | Qwen2.5-coder-7B STaR-SFT ( SFT+RL all reward)    | 65.5  |
>
> [1]: Using LLM to select the right SQL Query from candidates
> [2]: Omnisql: Synthesizing high-quality text-to-sql data at scale

---

### Official Review · Reviewer_yRRV · 2025-05-12

**Rating:** 7
**Confidence:** 4
**Ethics Flag:** 1

**Summary:**

Inspired by DeepSeek R1's reinforcement learning (RL) training using GRPO, this work investigates the application of RL to the Text-to-SQL task. The authors developed a reasoning-enhanced Text2SQL system on top of Qwen2.5-Coder and demonstrated its effectiveness through both quantitative evaluations—such as performance on the in-domain BIRD benchmark and out-of-domain Spider-related datasets—and qualitative analysis, highlighting emergent reasoning capabilities. Additionally, by integrating their RL-trained language model with the state-of-the-art CHASE framework, they achieved performance competitive with leading closed-source models on the Text2SQL task.

**Reasons To Accept:**

1. To further improve the performance, the authors designed multiple rewards, which are key innovations to make sure GRPO based RL training works for Text2SQL task. These rewards design can inspire others who also want to apply RL training to other tasks.
2. Tthe final RL trained model shows competitve performance as other systems that rely on closed-source LLMs, the gap between open-source and closed-source is even smaller.

**Reasons To Reject:**

1. Given this claim -- "our RL-trained 14B-parameter model significantly outperforms larger proprietary models, e.g. o3-mini by 4% and Gemini-1.5-Pro-002 by 3% on the BIRD benchmark", it is pretty misleading, based on Table 2, the base model (Qwen2.5-coder-14B) already outperforms o3-mini and Gemini-1.5-Pro-002. Their RL-trained model just shows further performance gains. However, the baseline setup for Qwen2.5-coder-14B is not clear, is it zero-shot or few-shot? if few-shot, static or dynamic exemplars? we should prepare solid baseline results to make sure readers really understand the original Text2SQL performance with Qwen2.5-coder-14B.
2. Table 1 shows FS EX result, it doesn't make sense to show this result gold schema input, since one of the reward design is for schema linking, and this is critical for Text2SQL, we should avoid showing this cheat setting to highlight the good results numbers.
3.  For Table 3, the paper only shows CHASE + their RL-trained Qwen2.5-coder, how about CHASE with original Qwen2.5-coder? if the performance is also comparable, does that mean their RL training cannot improve Qwen2.5-coder's upperbound performance for Text2SQL capability?

---

> ### Author Response · Authors · 2025-06-01
> **Authors' response**
>
> We would like to thank the reviewer for their constructive feedback. In response to your comments, we conducted the following experiments, which we hope adequately address your concerns regarding the paper:
>
> 1) To ensure a fair comparison, all performance results reported in Table 2 are obtained in the zero-shot chain-of-thought (CoT) setting, using user-defined instructions adapted from the prompt proposed by CHASE-SQL, one of the current state-of-the-art methods in text-to-SQL. While it is true that the base model demonstrates strong performance in its 14B version, the 7B version initially performs at 58.73, which is lower than both o3-mini and Gemini 1.5 Pro. However, after applying our proposed reinforcement learning (RL) training, the performance of the 7B model improves significantly to 64.01, outperforming all closed-source models except Gemini 2.0 Flash. This result supports our claim that the proposed reward design and training method effectively boost performance beyond models like o3-mini and Gemini 1.5 Pro. Additionally, for the results with the filtered schema setting, the base 14B model performs worse than o3-mini but after RL-training the model achieves 4% performance gain and outperformed all of the strong base models.
>
> 2) We agree with the reviewer that schema linking is one of the most challenging sub-tasks in the text-to-SQL pipeline. However, we would like to clarify that the purpose of including perfect schema linking in our study was to demonstrate the theoretical upper bound achievable by our method under ideal conditions. As shown in Table 1 (EX column), our results using an imperfect schema linking method already exhibit improved performance over the baselines, reinforcing the effectiveness of our overall reward design.
>
> 3) We would like to thank the reviewer for highlighting this important experimental consideration. To address your question, we conducted an experiment comparing CHASE + Qwen-7B base, the SFT baseline, and our proposed Reasoning-SQL method. The results show that Reasoning-SQL consistently outperforms the other approaches, primarily due to its higher average@k performance, as detailed in Appendix A.4.
>
> | Model                                        | Ex    |
> |----------------------------------------------|-------|
> | CHASE + Qwen 2.5 7B GRPO reasoning-sql       | 68.05 |
> | CHASE + Qwen 2.5 7B STaR SFT                 | 66.92 |
> | CHASE + QWen 2.5 7B base                     | 66.12 |

---

> > ### Comment · Reviewer_yRRV · 2025-06-02
> >
> > Thanks for the responses! My concerns got resolved, I would like to see this paper accepted. I also update my score to reflect my  final judgement.

---

### Official Review · Reviewer_ChoJ · 2025-05-12

**Rating:** 6
**Confidence:** 4
**Ethics Flag:** 1

**Summary:**

Reasoning-SQL introduces an RL framework for text2sql, leveraging tailored partial rewards (execution accuracy, syntax, schema linking, n-gram similarity, and llm-as-a-judge) to overcome reward sparsity. Using GRPO, it trains LLMs to generate more accurate SQL in comparison with larger proprietary models on benchmarks like BIRD and Spider. Authors also argue Reasoning-SQL is more cost-effective.

**Questions To Authors:**

See weekness.

**Reasons To Accept:**

This work presents a RL training paradigm with new reward design choices tailored specifically for SQL generation tasks. Unlike conventional RL approaches that primarily rely on execution accuracy, Reasoning-SQL introduces a composite reward and demonstrate promising results in this direction.

**Reasons To Reject:**

1. complex reward functions may lead models to exploit unintended behavior. Even though the authors study how each reward by itself perform, it doesn't seem the authors have provided evidence on the justification of the particular reward combination choice (i.e. ablation on different combinations of rewards is lacking).
2.  The paper introduces a composite reward system, the authors argue it's necessary to capture the multifaceted nature of SQL generation tasks. However, components like the llm-as-a-judge reward introduce uncertainty, which confilct with recent studies that advocate for simple, verifiable reward designs to enhance performance and mitigate reward hacking risks [1, 2]. Therefore, justification for using a composite reward should be rigorously evaluated against simpler alternatives. But this part of experiment is lacking.
3. The evaluation primarily benchmarks Reasoning-SQL against proprietary models. However, it omits comparisons with recent open-source models such as XiYanSQL [3], OmniSQL [4] and Think2SQL [5].
4. no comparison of PPO and GRPO results on the reward design choice.
5. Appendix A.4 indicates that Reasoning-SQL tends to produce less diverse SQL queries compared to the SFT baseline, with only a ~7% improvement from greedy sampling. This reduced diversity could pose challenges in applications requiring varied query formulations, potentially limiting the model's generalizability.

[1] Mroueh, Youssef. "Reinforcement Learning with Verifiable Rewards: GRPO's Effective Loss, Dynamics, and Success Amplification." arXiv preprint arXiv:2503.06639 (2025).
[2] Meng, F., et al. "Mm-eureka: Exploring visual aha moment with rule-based large-scale reinforcement learning." arXiv preprint arXiv:2503.07365 (2025).
[3] Gao, Yingqi, et al. "Xiyan-sql: A multi-generator ensemble framework for text-to-sql." arXiv preprint arXiv:2411.08599 (2024).
[4] Li, Haoyang, et al. "Omnisql: Synthesizing high-quality text-to-sql data at scale." arXiv preprint arXiv:2503.02240 (2025).
[5] Papicchio, Simone, et al. "Think2SQL: Reinforce LLM Reasoning Capabilities for Text2SQL." arXiv preprint arXiv:2504.15077 (2025).

---

> ### Author Response · Authors · 2025-06-01
> **Authors' response (part 1)**
>
> We would like to thank the reviewer for their constructive feedback. In response to your comments, we conducted the following experiments, which we hope adequately address your concerns regarding the paper:
>
> 1) The reward design proposed in our work builds upon valuable insights from prior studies in the text-to-SQL domain. As highlighted in previous works such as DIN-SQL [1], codeS [2], and especially MapleRepair [3], errors in text-to-SQL generation are typically categorized into syntax errors, schema errors, semantic errors, logic errors, convention errors, and gold SQL type errors.
> Our reward components are specifically designed to target each of these error categories, assigning partial rewards based on the model’s correctness in each area. More specifically: The syntax check reward is designed to reduce syntax errors. The schema linking reward encourages accurate schema references, which account for a significant proportion of text-to-SQL errors. For semantic and logical errors, which require reasoning, we employ the LLM-as-a-judge reward. To address convention errors, our n-gram-based reward helps the model learn and adhere to standard SQL conventions. Although Table 1 of the paper includes an ablation study of the reward components, we have now added two additional experiments. In these, we trained the Qwen-coder 7B model using only the n-gram reward and only the schema linking reward, respectively. These experiments, presented below, further demonstrate the effectiveness of our proposed reward combination in improving overall model performance.
>
> | Model                               | Ex    | FS Ex  |
> |-------------------------------------|-------|--------|
> | Qwen-7B-coder (exe + syn)           | 62.84 | 68.97  |
> | Qwen-7B-coder (exe + schema)        | 63.10 | 69.03  |
> | Qwen-7B-coder (exe + ngram)         | 63.03 | 68.90  |
> | Qwen-7B-coder (llm-as-a-judge)      | 63.75 | 69.16  |
> | Qwen-7B-coder (all-rewards)           | 64.01 | 70.66  |
>
> 2) We agree with the reviewer that simpler rewards are generally harder to hack. However, the simplest reward design in the text-to-SQL domain—execution accuracy—has a significant limitation, as discussed in the paper. Its binary nature fails to provide informative feedback when a query is only partially correct. This limitation is particularly critical for policy optimization methods like GRPO, which depend on group-based reward signals. In cases where all queries receive either zero or one as a reward, the model receives little to no gradient signal for learning and improvement. Furthermore, we would like to highlight that Table 1 in our paper presents a direct comparison between execution accuracy as the simplest reward and our proposed composite reward design. Our approach achieves an execution accuracy of 64.01%, compared to 62.32% achieved by using execution accuracy alone, demonstrating the added value of our more nuanced reward strategy.
>
> 3) We would like to thank the reviewer for highlighting recent works in the field. We note, however, that a direct comparison with XiYan-SQL is already included in Table 3 of our paper. As for the Omni-SQL paper, it was released concurrently with our submission, and ThinkSQL was published after both our submission and the COLM conference deadline, making it impossible to include it in the original paper. Nevertheless, as per your suggestion, we included a direct comparison with both Omni-SQL and ThinkSQL on the BIRD development set, as shown in the following table, where our approach significantly outperform both of them:
> | Method              | Execution acc |
> |---------------------|---------------|
> | Think2SQL 7B        | 56.1          |
> | Omni-SQL 32B        | 69.23         |
> | Reasoning-SQL 14B   | 72.29         |
>
> 4) The primary goal of our approach in this paper is to propose a reward design strategy aimed at improving the model’s reasoning ability and execution accuracy, rather than to compare different policy optimization methods. We chose to use GRPO primarily because it has a lower memory footprint compared to PPO and has demonstrated strong performance on reasoning-intensive tasks such as MATH and code generation.

---

> > ### Comment · Reviewer_ChoJ · 2025-06-09
> >
> > I would like to thank the authors' for your answers and additional experiments on reward design choices. They have addressed some of my concerns and I raised the rating.

---

> ### Author Response · Authors · 2025-06-01
> **Authors' response (part 2)**
>
> 5) As noted in the DeepSeek R1 [5] paper and other recent works [4], training with GRPO does not improve the model’s pass@k performance beyond its pre-trained baseline. However, it can enhance average@k or majority@k performance by improving pass@1 and the overall quality of generated responses. While this may slightly reduce the model's theoretical upper bound (i.e., maximum possible pass@k), in practice, achieving that upper bound is highly challenging. In contrast, improving the average quality of responses is often more valuable—especially for recent text-to-SQL methods like XiYan-SQL and CHASE-SQL, which rely on query selection from a candidate pool. In these cases, the quality of that candidate pool (average@k) is crucial. This benefit is evident when comparing the performance of Reasoning-SQL 7B and the SFT baseline within the CHASE-SQL pipeline, as demonstrated below. Further supporting evidence can also be found in Appendix A.4, where the RL-trained model shows improved average@k performance.
>
> | Model                                        | Ex    |
> |----------------------------------------------|-------|
> | CHASE + Qwen 2.5 7B GRPO reasoning-sql       | 68.05 |
> | CHASE + Qwen 2.5 7B STaR SFT                 | 66.92 |
> | CHASE + QWen 2.5 7B base                     | 66.12 |
>
> [1]: Din-sql: Decomposed in-context learning of text-to-sql with self-correction
> [2]: Codes: Towards building open-source language models for text-to-sql
> [3]: A Study of In-Context-Learning-Based Text-to-SQL Errors
> [4]: Does Reinforcement Learning Really Incentivize Reasoning Capacity in LLMs Beyond the Base Model?
> [5]: Deepseek-r1: Incentivizing reasoning capability in llms via reinforcement learning

---

> ### Author Response · Authors · 2025-06-07
> **Reminder**
>
> We’re writing to kindly follow up on our responses to your valuable comments. If you haven’t had a chance yet, we would greatly appreciate it if you could take a moment to review our responses and, if appropriate, update your scores accordingly.
>
> Thank you again for your time and valuable feedback throughout this process. Please don’t hesitate to let us know if any further clarification is needed from our side.

---

> > ### Comment · Area_Chair_PhCD · 2025-06-07
> > **Response to Author Rebuttal**
> >
> > Gentle reminder to please respond to the author's rebuttal

---

> > > ### Comment · Reviewer_ChoJ · 2025-06-09
> > >
> > > Thanks for the reminder! The authors' rebuttal has answered my questions and addressed some of my concerns. I raised the rating 6.

---

### Decision · Program_Chairs · 2025-07-08

**Decision:**

Accept

**Comment:**

The authors present a RL framework for text2SQL generation that employs a composite reward system (using execution accuracy, syntax checking, etc.) to address reward sparsity. The authors show competitive performance against larger models and also show emergent reasoning capabilities while also being cost sensitive.

Reviewers had initially raised concerns about (i) the justification for the composite reward design (ii) misleading performance claims and experimental setup clarity (iii) requirement of gold answers (iv) marginal reward improvement and training stability — but the authors were able to satisfactorily address a large number of concerns.

Overall, the paper makes good technical contributions with the composite design and empirical results demonstrating incremental progress with practical applications.